# Production of Foot-and-Mouth Disease Type O and A Vaccine Antigens on a Pilot Scale and Determination of Optimal Amount of Antigen for Monovalent Vaccines

**DOI:** 10.3390/vaccines11071156

**Published:** 2023-06-26

**Authors:** Jae Young Kim, Ji-Hye Lee, Jong Min Yang, Seo-Yong Lee, Sun Young Park, Jong Sook Jin, Dohyun Kim, Jung-Won Park, Jong-Hyeon Park, Sang Hyun Park, Young-Joon Ko

**Affiliations:** 1Animal and Plant Quarantine Agency, Gimcheon-si 39660, Republic of Korea; ivorikim@korea.kr (J.Y.K.); in75724@korea.kr (J.S.J.);; 2Department of Research and Development, FVC, Gimcheon-si 39660, Republic of Korea; 3Department of Development, SNC Bio, Hanam-si 12930, Republic of Korea

**Keywords:** FMDV, vaccine, monovalent, scale-up

## Abstract

Foot-and-mouth disease (FMD) is a highly infectious disease affecting cloven-hoofed animals and causes significant economic losses to the livestock industry. The Type O PanAsia-2 (O PA-2) vaccine strain is protective against a wide range of serotype O FMD virus (FMDV) strains in East Asia, and A22 Iraq/24/64 (A22 IRQ) is the most widely used vaccine strain in FMD vaccine antigen banks. The aim of this study was to produce antigens from O PA-2 and A22 IRQ viruses using a 100 L bioreactor and evaluate the protective efficacy of varying antigen concentrations in pigs. More than 2 μg/mL of the antigen was recovered from the O PA-2 and A22 IRQ virus-infected supernatants. Further, inactivation of O PA-2 and A22 IRQ by binary ethyleneimine revealed that the viral titers decreased below 10^−7^ TCID_50_/mL within 13 h and 9 h, respectively. The O PA-2 and A22 IRQ vaccines, containing 10 μg and 5 μg of antigen, respectively, provided protection against homologous viruses in pigs. This is the first report demonstrating that the antigens obtained from the pilot-scale production of O PA-2 and A22 IRQ are viable candidate vaccines. These results will pave the way for industrial-scale FMD vaccine production in South Korea.

## 1. Introduction

Foot-and-mouth disease (FMD) is a highly contagious viral disease that affects cloven-hoofed animals and leads to significant economic damage to the livestock sector [1]. The causative agent responsible for FMD is the FMD virus (FMDV), which belongs to the *Aphthovirus* genus within the *Picornaviridae* family. This virus encompasses seven recognized serotypes: O, A, Asia 1, C, South African Territories (SAT) 1, SAT 2, and SAT 3 [2]. The viral genome consists of single-stranded positive RNA and encodes four structural proteins (VP1, VP2, VP3, and VP4) as well as eight nonstructural proteins (L, 2A, 2B, 2C, 3A, 3B, 3C, and 3D) [3].

To protect livestock from FMD, many countries worldwide have adopted FMD vaccination policies [4]. In the Republic of Korea, FMD outbreaks occurred in 2000 and 2002, followed by a large-scale outbreak in 2010, and a vaccination policy has been implemented throughout the country since December 2020. However, there is no domestically produced FMD vaccine, and all vaccines have been imported.

The Type O PanAsia-2 (O PA-2) vaccine strain is highly efficacious against several FMDV topotypes in East Asia [5]. A22 Iraq/24/64 (A22 IRQ) is a widely used vaccine strain included in most FMD vaccine antigen banks [6]. Since there have been outbreaks of types O and A in South Korea, we developed O PA-2 and A22 IRQ vaccine strains using genetic recombination. Trial vaccines with O PA-2 and A22 IRQ have already been developed and evaluated [7,8]. However, these experiments were conducted using antigens prepared on a small scale using flasks to amplify FMDV. Therefore, the objective of this study was to produce O PA-2 and A22 IRQ antigens on a pilot scale using a 100 L bioreactor and determine the optimal amount of antigens that can protect pigs against FMDV. 

## 2. Materials and Methods

### 2.1. Cells and Viruses

The BHK-21 suspension cells—developed from the original adherent cell line, BHK-21 [C-13] (ATCC, Manassas, VA, USA), by the Animal and Plant Quarantine Agency (APQA) and the Korea Research Institute of Bioscience & Biotechnology for use in suspension culture with serum-free media—were adapted for growth in Cellvento^TM^ BHK-200 cell culture medium (Merck, Darmstadt, Germany) by incubation at 110 rpm in a shaking incubator at 37 °C with 5% CO_2_. The cell number and viability were assessed using the trypan blue exclusion method employing an automated cell counter (Vi-Cell XR, manufactured by Beckman Coulter Inc., Brea, CA, USA). To cultivate the cells, a cell suspension containing 3 × 10^5^ cells/mL was grown in 70% of the total medium volume for 3.5 days until it reached approximately 3 × 10^6^ cells/mL. Then, without removing the spent medium, 30% of fresh Cellvento medium was added. Subsequently, the FMDV was introduced into the flasks or bioreactor.

The O PA-2 and A22 IRQ strains were obtained from O PAK/44/2008 (GenBank accession no. GU384682) and A22 Iraq/24/64 (GenBank accession no. AY593764.1), respectively. The two FMD vaccine strains were recombinant viruses in which P1 of the recombinant O1 Manisa backbone was replaced with P1 of O/PAK/44/2008 or A22/Iraq/24/64 and termed O PA-2 and A22 IRQ, respectively [7]. FMD vaccine viruses were inoculated into BHK-21 suspension cells, and vaccine antigens were produced under optimized production conditions. The O PA-2 was inoculated at a multiplicity of infection (MOI) of 0.001, and the virus was harvested 16 h post infection (hpi). A22 IRQ was inoculated at an MOI of 0.001, and the virus was collected at 24 hpi. The viral infection supernatants were collected by centrifugation (4000× *g*, 20 min, 4 °C). These supernatants were used to quantify FMDV particles (146S) and viral titers.

### 2.2. Pilot-Scale Production of FMD Vaccine Antigens

Cellvento BHK-200 cell culture medium was prepared using a mixer (Sartorius, Goettingen, Germany). A total of 70 L of the medium was then transferred to a 100 L bioreactor (Sartorius) with an initial cell density of 3 × 10^5^ cells/mL. The bioreactor was equipped with probes to monitor and regulate the temperature and dissolved oxygen levels, maintaining them at 37 °C and 50% air saturation, respectively. The pH was controlled within the range of 7.2–7.4 by adding CO_2_ and a 1 M NaOH solution. The agitation speed was set at 100 rpm. Samples were collected every 24 h to assess the cell density and viability. When the cell density reached approximately 3 × 10^6^ cells/mL, 30 L of fresh Cellvento BHK-200 cell culture medium was added to the bioreactor. Subsequently, O PA-2 or A22 IRQ was introduced to the bioreactor along with fresh medium at MOIs of 0.001 and 0.05, respectively. The supernatants infected with O PA-2 and A22 IRQ were collected at 16 and 24 hpi, respectively. These supernatants were then transferred to a 250 L inactivation tank (BT Resources, Gyeonggi, Korea) and inactivated using 3 mM binary ethyleneimine (BEI, Sigma-Aldrich, St. Louis, MO, USA). The inactivation process took place for 24 h at 26 °C, employing two individual tanks. Hourly samples were collected from the inactivation tank during the 0 to 6 h period to evaluate the inactivation kinetics of each strain. The remaining BEI was neutralized by adding 10% of a 1 M sodium thiosulfate solution (Daejung Chemicals, Siheung-si, Korea), resulting in a final concentration of 2%. The inactivated viruses were treated with 50% Polyethylene Glycol (PEG) 6000 (Sigma-Aldrich) solution into a final 7.5% concentration of PEG. The solution was added with NaCl (0.5 M) and then incubated for 16 h at 4 °C. Subsequently, they were harvested using a continuous centrifuge (Tomoe, Tokyo, Japan) operating at 16,400 rpm and a flow rate of 42 L/h. The resulting pellet was resuspended in 50 mM Tris containing 300 mM KCl (pH 7.6) and stored at −70 °C until further use.

### 2.3. Virus Titration

Viral titers were determined using adherent BHK-21 cells via endpoint titration according to the Spearman-Kärber method, which was expressed as the tissue culture infectious dose affecting 50% of the culture (TCID_50_) per milliliter [9].

### 2.4. Quantification of FMDV Particles

The supernatant was mixed with chloroform (Merck KGaA, Darmstadt, Germany) in a 1:1 ratio (*v*/*v*) by vigorous inversion for 5 min. After mixing, the mixture was centrifuged at 3000× *g* for 15 min at 4 °C, and the aqueous phase located above the organic solvent was collected. This process was repeated twice, and the resulting samples were centrifuged at 16,000× *g* for 10 min to obtain the supernatants. The samples were then treated with benzonase (Sigma-Aldrich) at a final concentration of 0.025 units/μL and incubated at 37 °C with shaking for 1 h. After the incubation, the samples were centrifuged at 16,000× *g* for 10 min at 4 °C to obtain a clear sample. To further refine the samples, they were filtered through 0.22 μm Millex-GV filters (Merck KGaA) and loaded onto a size-exclusion high-performance liquid chromatography (SE-HPLC) column for quantification of 146S intact virus particles. The SE-HPLC analysis utilized a TSKgel G4000PWXL (300 × 7.8 mm) column (TOSOH Bioscience, Tokyo, Japan) combined with a TSKgel PWXL GuardCol (40 × 6.0 mm) guard column (TOSOH Bioscience) using an Agilent 1260 Infinity II system (Agilent Technologies, Santa Clara, CA, USA). The mobile phase consisted of 30 mM Tris-HCl and 400 mM NaCl at pH 8.0, and the flow rate was set at 0.5 mL/min. The area under the target peak was integrated using the OpenLAB chromatography data system (CDS) ChemStation Edition Rev. C01.10, and the amount of 146S antigen (g/mL) was calculated based on a previous study [10].

### 2.5. Inactivation Kinetics

BEI was prepared by dissolving BEA (bromoethylamine hydrobromide) in 10 mL of a solution of 0.2 N sodium hydroxide (Sigma-Aldrich, St. Louis, MO, USA) to achieve a concentration of 0.1 M. The solution was placed in a shaking incubator at 37 °C and 100 rpm for 1 h. The pH of the solution was adjusted to a range of 8.5–9. A fresh solution was prepared prior to each use. Samples were collected every hour from 0 to 6 h from the inactivation tank to assess the virus inactivation rate for each strain. The graph for each strain was extrapolated as a linear line to analyze the kinetics of FMDV inactivation. The remaining BEI was neutralized with 10% of the volume using 1 M sodium thiosulfate to a final concentration of 2%.

### 2.6. Transmission Electron Microscopy

The inactivated antigen, concentrated using Polyethylene Glycol (PEG), was carefully placed on sucrose gradients (15–45%) and subjected to ultracentrifugation at 100,000× *g* for a duration of 4 h. The band located between the sucrose layers of 30% and 35% was collected and further subjected to ultracentrifugation at 100,000× *g* for 4 h. The resulting pellet was resuspended again and subjected to dialysis at 4 °C against a solution of 50 mM Tris and 300 mM KCl (pH 7.6) to eliminate any residual sucrose. A small amount of the purified FMDV suspension was applied to formvar-coated grids and negatively stained with 1% uranyl acetate. Finally, the FMDV particles were observed using a transmission electron microscope (H-7100FA; Hitachi, Tokyo, Japan).

### 2.7. Protective Efficacy of Monovalent Vaccines in Pigs

The antigens O PA-2 and A22 IRQ, as described in Section 2.2 without further purification, were mixed with 1% saponin (Sigma-Aldrich) and 10% aluminum hydroxide gel (General Chemical, NJ, USA) to prepare a monovalent vaccine. Subsequently, the pre-warmed ISA 206 VG adjuvant (Seppic, Paris, France) was introduced in a 1:1 ratio, resulting in a 1 mL dose for type O or A. The mixtures were subjected to incubation at 20 °C for 1 h, in a light-restricted water bath, and subsequently stored at 4 °C until required. For the O PA-2 vaccine, a total of 17 two-month-old pigs, which had not previously received any FMD vaccination, were segregated into four groups. Pigs in groups 1–3 (*n* = 5 for all groups) were immunized with a monovalent O PA-2 vaccine (1 mL) containing 20, 15, and 10 µg of antigen, respectively. The control group consisted of two unvaccinated pigs. Each group of pigs was intramuscularly administered vaccine containing various payloads of antigens followed by homologous virus challenge using O PA-2 (1 × 10^5^ TCID_50_/0.1 mL) in the heel bulb at 28 days post vaccination (dpv). Blood samples were collected at 0, 7, 14, 21, and 28 dpv. During the experiment, one pig each from groups 2 and 3 died of pleuropneumonia and Glasser’s disease, respectively. For the A22 IRQ vaccine, the same experiment was performed with the same number of pigs as described above. However, the virus challenge was conducted with A22 IRQ. If pigs showed clinical signs of FMD, they were immediately isolated to prevent further virus challenge to the pigs not showing clinical signs. The clinical score was calculated by the following parameters: the presence of vesicles on the foot, which was contingent upon the number of affected feet (up to a maximum of 4 points); and observable mouth lesions on the tongue (1 point), gums and/or lips (1 point), or snout (1 point) (maximum score = 3). Protection was defined as all regions of the pigs, except for the injection site, showed no clinical signs throughout the experimental period, within 7 days following viral challenge. Clinical scores were calculated by summing the scores corresponding to FMD lesions or signs (maximum score = 7) [11].

### 2.8. Virus Neutralization Test

The virus neutralization (VN) test was conducted in accordance with the procedures outlined in the WOAH terrestrial manual [12]. Prior to testing, the sera were subjected to inactivation at 56 °C for a duration of 30 min. Fifty microliters of sera, initially diluted at a 1/4 ratio, were mixed with 50 μL of each virus containing 100 TCID_50_. Following an incubation for 1 h at a temperature of 37 °C, 50 μL of LFPK cells (fetal porcine kidney cells; supplied by Plum Island Animal Disease Center, Orient, NY, USA) (0.5 × 10^6^ cells/mL) was added to each well. The plates were sealed and incubated at 37 °C with 5% CO_2_ for 2–3 days. The VN titer was determined using the Spearman-Kärber method and reported as a log_10_ value.

### 2.9. Statistical Analysis

We produced FMD vaccine antigens in a 100 L bioreactor in triplicate. Virus titration, quantification of 146S particles, and inactivation kinetics were performed three times, and the mean and standard deviation of all values were reported. VN titers and inactivation kinetics were analyzed using GraphPad Prism version 9 (GraphPad Software, San Diego, CA, USA) for visual representations.

## 3. Results

### 3.1. Scaling up Production of FMD Vaccine Antigen

The O PA-2 and A22 IRQ were propagated in a 100 L bioreactor based on the optimal conditions determined from the experiments in the flask. The O PA-2 and A22 IRQ viruses were inoculated into suspension cells at an MOI of 0.001, and the viral infection supernatants were collected at 16 h and 24 h, respectively. The virus titers of the supernatants from O PA-2 and A22 IRQ were 2.21 × 10^7^ TCID_50_/mL and 1.71 × 10^7^ TCID_50_/mL, respectively (Figure 1a,b). The amount of 146S antigen in each supernatant was 3.6 µg/mL and 2.0 µg/mL for O PA-2 and A22 IRQ, respectively. The inactivation kinetics after BEI treatment showed a linear slope, and the titers of O PA-2 and A22 IRQ viruses decreased to below 10^−7^ TCID_50_/mL within 13 h and 9 h, respectively (Figure 1c,d). After virus inactivation, the physical integrity of the viruses during scale-up in a 100 L bioreactor was examined under an electron microscope, which revealed FMDV particles with a diameter of 25–30 nm (Figure 1e,f).

### 3.2. Immunogenicity and Protective Efficacy of O PA-2 Vaccine in Pigs

The optimal amount of O PA-2 antigen was determined by preparing trial vaccine doses using 20 µg, 15 µg, and 10 µg antigen per dose, which was administered to 2-month-old growing pigs. VN titers were measured for up to 7 days after the virus challenge. Although all three groups showed more than 1:45 (cutoff of 1.65 log_10_) VN titers, starting from 21 dpv (Figure 2a–c), the VN titers were less than 1:10 at the time of virus challenge in the negative control group (Figure 2d). As shown in Table 1, no clinical symptoms were observed in the groups immunized with the monovalent vaccine using the O PA-2 antigen, whereas pigs in the negative control group showed typical clinical signs of FMD (Table 1).

### 3.3. Immunogenicity and Protective Efficacy of A22 IRQ Vaccine in Pigs

The optimal amount of A22 IRQ antigen was determined by preparing trial vaccines using 15 µg, 10 µg, and 5 µg antigen per dose and administered to 2-month-old growing pigs (*n* = 5). Similar to the O PA-2 vaccine, pigs in groups 1–3 (all with *n* = 5) were immunized with monovalent A22 IRQ vaccine (1 mL) containing 15, 10, and 5 µg of antigen, respectively. Pigs in all groups were challenged at 28 dpv with A22 IRQ. VN titers were measured for up to 7 days after the virus challenge. In the 15 µg antigen group, all pigs had more than a 1.65 log_10_ VN titer at 28 dpv. On the other hand, in the 10 µg and 5 µg antigen groups, there was one pig each with a VN titer less than 1.65 log_10_ on day 28 (Figure 3b,c). Two pigs did not show VN titers above 1.65 log_10_ in the negative control group at the time of virus challenge. No clinical signs were observed in the monovalent vaccination group with antigens from A22 IRQ, whereas pigs in the negative control group showed typical clinical signs of FMD (Table 2).

## 4. Discussion

Based on a previous report [6] that FMD type O is the most common type of FMD worldwide and that inactivated vaccines made from O PA-2 have good vaccine matching against various FMDV in Asia, O PA-2 was produced by genetic recombination to be used as a vaccine [7]. In addition, A22 IRQ has been shown to protect against challenge with many heterologous A strains [6]. For these reasons, recombinant A22 IRQ was produced for use as a vaccine strain [7].

Previously, we evaluated the protective efficacy of a trivalent test vaccine containing O PA-2 and A22 IRQ and a monovalent test vaccine containing O PA-2 [7,8]. However, these experiments were performed using antigens produced in small-scale flask cultures.

Generally, FMD vaccine factories manufacture FMD vaccine antigen on a large scale using multiple bioreactors. Industrial-scale production of FMD vaccine requires huge experimental costs, which makes comprehensive process optimization very difficult. Therefore, a scaled-down model representative of industrial FMD vaccine production was required. Therefore, in this study, we scaled up production of FMD O PA-2 and A22 IRQ vaccine antigens at the 100 L level as a step prior to industrial-scale production in an FMD vaccine factory, and produced trial monovalent vaccines using these antigens to evaluate immunogenicity and protective efficacy in pigs. It was previously reported that a 14 L bioreactor can serve as a scaled-down model for the 4000 L cell culture process [13].

When O PA-2 and A22 IRQ were grown in a 100 L bioreactor, viral titers were similar, but antigen concentrations were different in the virus infection supernatant. The exact cause is unknown, but myristoylation of the VP4 N-terminus is known to be involved in the mature cleavage of VP0 into VP4 and VP2 during assembly of the viral particle 146S, which allows RNA to enter the viral particle [14]. Further study on the relationship between VP4 myristoylation and the amount of FMD vaccine antigen is warranted. In this study, we were able to recover more than 2.0 µg/mL of antigen from the virus infection supernatant using a 100 L bioreactor. This is consistent with previous reports where about 1–3 µg/mL of antigen was recovered from the virus infection supernatant [15,16,17,18].

The quantification of 146S particles can be achieved through Sucrose density gradient ultracentrifugation or SE-HPLC. Sucrose density gradient ultracentrifugation involves multiple labor-intensive steps, such as the preparation of sucrose gradient tubes, ultracentrifugation, and manual measurement of peak areas. Moreover, this method has limitations in terms of the number of samples that can be processed concurrently. To overcome these limitations, SE-HPLC was introduced as a replacement for the traditional Sucrose density gradient method [10]. Consequently, in this study, SE-HPLC was utilized to quantify the 146S antigen. A strong correlation was established between SE-HPLC and sucrose gradient centrifugation in the quantification of FMDV particles [10].

In-process quality control of FMDV inactivation kinetics was performed by extrapolation to ensure less than 1 infectious particle per 10^4^ L of liquid formulation at the end of the inactivation period [12]. Although the optimal BEI inactivation of O PA-2 and A22 IRQ in a preliminary study using flask culture was 2 mM at 26 °C, we treated O PA-2 and A22 IRQ with 3 mM BEI on a 100 L scale to inactivate FMDV completely. The virus titers decreased steadily until 6 h after BEI treatment and dropped to the threshold of −7 log TCID_50_/mL within 16 h post BEI treatment [12].

Although the amount of the 146S antigen per vaccine dose normally ranges from 1 to 10 µg [19], a previous report has shown that antigen doses of 10 µg or more reliably induce high VN titers and provide high protection against both homologous and heterologous FMDV in pigs [20]. In addition, an antigenic dose of 15 µg was used for O PA-2 in a previous study [8]. In this study, the amount of antigen was adjusted from 10 µg to 20 µg to investigate the protective efficacy in pigs. Even at the lowest concentration of 10 µg, no clinical signs were observed in pigs, indicating that a single 10 µg dose of the antigen was sufficient for protection against FMDV O PA-2.

Similarly, the antigen dose range of A22 IRQ was adjusted from 5 to 15 µg based on a previous report that more type O antigen is required than type A to achieve an equivalent potency [19]. In this study, the antigenic payload of 5 µg per dose showed complete protection against FMDV. Some pigs (No. 21 and No. 26) had VN titers below the threshold (1.65 log_10_) at the time of virus challenge but had no clinical signs, indicating that they were still protected from the virus. Our results corroborate with those of previous reports showing that some animals can be protected with lower levels of VN titers [6].

In conclusion, this is the first report demonstrating a pilot-scale production of recombinant O PA-2 and A22 IRQ antigens that can protect pigs against virus challenge at dosages of 10 µg and 5 µg, respectively. These results will pave the way for industrial-scale FMD vaccine production in South Korea.

## Figures and Tables

**Figure 1 vaccines-11-01156-f001:**
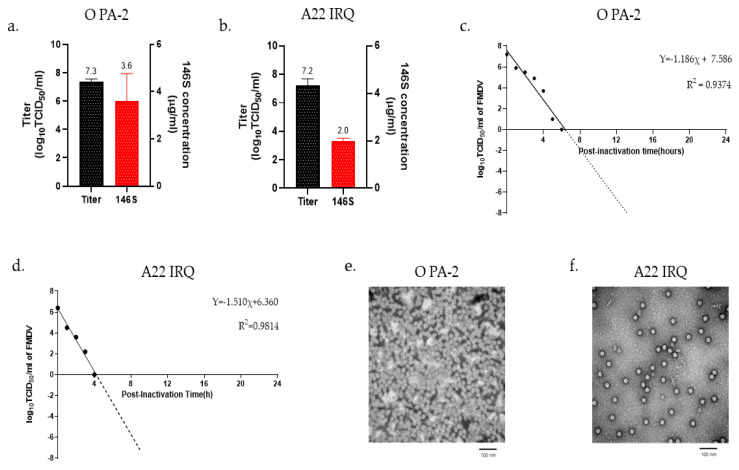
Antigen production was carried out in a 100 L bioreactor. (**a**,**b**) The quantity of 146S antigen and virus titer in the supernatant resulting from virus infection in the bioreactor. (**c**,**d**) The virus inactivation kinetics after treatment with binary ethyleneimine (BEI) was also examined. The supernatant obtained from virus infection was subjected to two treatments with 3 mM BEI (at 0 h and 16 h) and allowed to incubate for a total of 24 h. Samples were collected hourly from 0 to 6 h to determine the kinetics of virus inactivation for each virus strain. The resulting data were extrapolated and plotted as a linear line to analyze the inactivation kinetics of FMDV. (**e**,**f**) The inactivated 146S antigen was concentrated through PEG precipitation, purified via sucrose gradient ultracentrifugation, and observed using a transmission electron microscope (TEM).

**Figure 2 vaccines-11-01156-f002:**
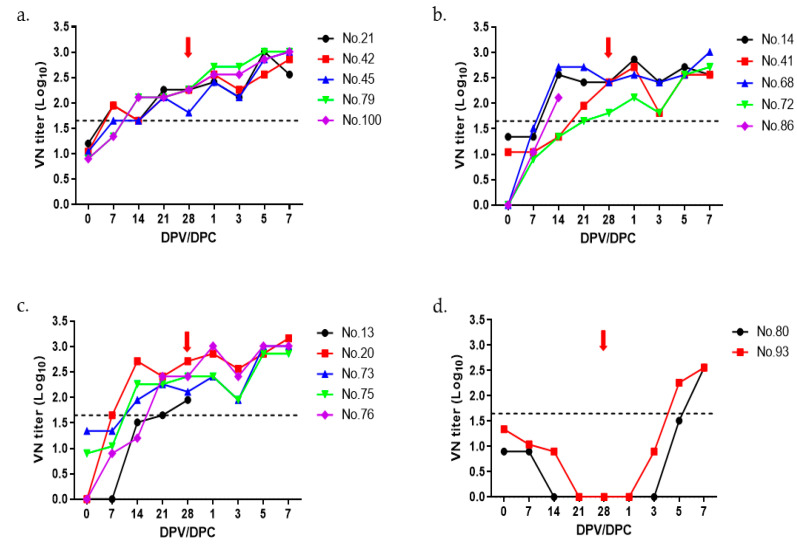
The titers of virus neutralization in pigs following immunization with the O PA-2 vaccine. VN tests were conducted on sera collected from pigs (5 pigs per group except for negative control with 2 pigs) that received trial vaccines containing four different antigen payloads of 20 µg (**a**), 15 µg (**b**), 10 µg (**c**), and negative control (**d**) up to 28 days post vaccination (dpv). Blood samples were obtained on days 0, 7, 14, 21, and 28 dpv. At 28 dpv, the pigs were exposed to O PA-2 with a viral dose of 1 × 10^5^ TCID_50_/0.1 mL in the heel bulb. The red arrows indicate the virus challenge introduced at 28 dpv. The dotted line represents the VN titer of 1:45 (1.65 log_10_), which is considered positive according to the guidelines in the WOAH terrestrial manual [12].

**Figure 3 vaccines-11-01156-f003:**
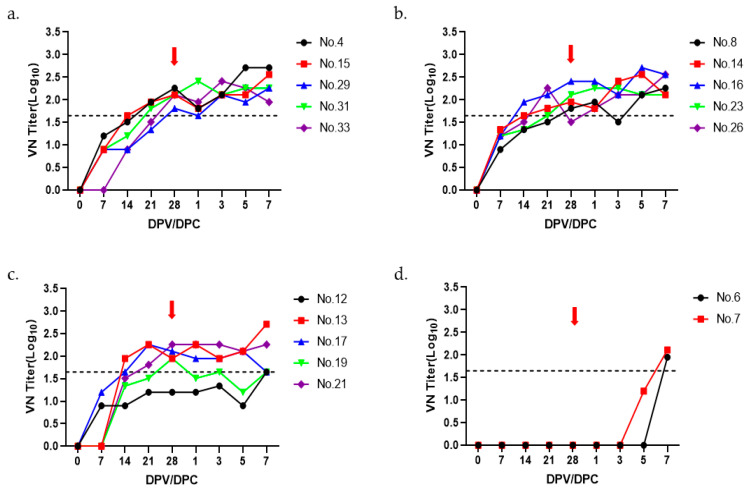
The virus-neutralizing titers observed in pigs after immunization with the A22 IRQ vaccine. The pigs (*n* = 17) were divided into four groups (5 pigs per group except for negative control with 2 pigs), receiving doses of 15 µg (**a**), 10 µg (**b**), and 5 µg (**c**) and negative control (**d**). Blood samples were collected at various time points: 0, 7, 14, 21, and 28 days post vaccination. At 28 days post vaccination, the pigs were challenged with A22 IRQ, administered at a viral dose of 1 × 10^5^ TCID_50_/0.1 mL in the heel bulb. Blood samples were further collected up to 7 days post challenge. The red arrows indicate the virus challenge introduced at 28 days post vaccination. The dotted line on the graph represents the VN titer threshold of 1:45 (1.65 log_10_), as deemed positive in accordance with the guidelines presented in the WOAH terrestrial manual [12].

**Table 1 vaccines-11-01156-t001:** Summary of immune response and protective efficacy of the O PA-2 monovalent vaccine in pigs.

Group	Pig ID	Virus-Neutralizing Titer					Clinical Score	Protection
		0 dpv	7 dpv	14 dpv	21 dpv	28 dpv		
20 µg	#21	1.20	1.95	1.65	2.26	2.26	0	Yes
	#42	1.04	1.95	1.65	2.11	2.26	0	Yes
	#45	1.04	1.65	1.65	2.11	1.81	0	Yes
	#79	0.9	1.34	2.11	2.11	2.26	0	Yes
	#100	0.9	1.34	2.11	2.11	2.26	0	Yes
15 µg	#14	1.34	1.34	2.56	2.41	2.41	0	Yes
	#41	1.04	1.04	1.34	1.95	2.41	0	Yes
	#68	<0.9	1.51	2.71	2.71	2.41	0	Yes
	#72	<0.9	0.9	1.34	1.65	1.81	0	Yes
	#86	<0.9	1.04	2.11	-	-	-	-
10 µg	#13	<0.9	<0.9	1.51	1.65	1.95	-	-
	#20	<0.9	1.65	2.71	2.41	2.71	0	Yes
	#73	1.34	1.34	1.95	2.26	2.11	0	Yes
	#75	0.9	1.04	2.26	2.26	2.41	0	Yes
	#76	<0.9	0.9	1.20	2.41	2.41	0	Yes
Control	#80	0.9	0.9	<0.9	<0.9	<0.9	7	No
	#93	1.34	1.04	0.9	<0.9	<0.9	5	No

**Table 2 vaccines-11-01156-t002:** Summary of immune response and protective efficacy of the A22 IRQ monovalent vaccine in pigs.

Group	Pig ID	Virus-Neutralizing Titer					Clinical Score	Protection
		0 dpv	7 dpv	14 dpv	21 dpv	28 dpv		
15 µg	#4	<0.9	1.20	1.51	1.95	2.26	0	Yes
	#15	<0.9	0.9	1.65	1.95	2.11	0	Yes
	#29	<0.9	0.9	0.9	1.34	1.81	0	Yes
	#31	<0.9	0.9	1.20	1.81	2.11	0	Yes
	#33	<0.9	<0.9	0.9	1.51	2.11	0	Yes
10 µg	#8	<0.9	0.9	1.34	1.51	1.81	0	Yes
	#14	<0.9	1.34	1.65	1.81	1.95	0	Yes
	#16	<0.9	1.20	1.95	2.11	2.41	0	Yes
	#23	<0.9	1.20	1.34	1.65	2.11	0	Yes
	#26	<0.9	1.20	1.51	2.26	1.51	0	Yes
5 µg	#12	<0.9	0.9	0.9	1.20	1.20	0	Yes
	#13	<0.9	<0.9	1.95	2.26	1.95	0	Yes
	#17	<0.9	1.20	1.65	2.26	2.11	0	Yes
	#19	<0.9	<0.9	1.34	1.51	1.95	0	Yes
	#21	<0.9	<0.9	1.51	1.81	2.26	0	Yes
Control	#6	<0.9	<0.9	<0.9	<0.9	<0.9	6	No
	#7	<0.9	<0.9	<0.9	<0.9	<0.9	4	No

## Data Availability

No new data were created or analyzed in this study. Data sharing is not applicable to this article.

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
