# Peer review of "Production of Foot-and-Mouth Disease Type O and A Vaccine Antigens on a Pilot Scale and Determination of Optimal Amount of Antigen for Monovalent Vaccines"

_vaccines, 2023, doi:10.3390/vaccines11071156_

Round 1
Reviewer 1 Report
There is a lot of very instructive applied FMD vaccine research coming out of South Korea lately. This manuscript is another good example. It may not be highly innovative, but it is solid work that can be very useful to the field. However, the description of the methods could be improved.
Line 52: This is quite vague. There are many kinds of BHK-21 suspension cells. Which cell line was used? Is it available from a public repository such as ATCC or ECACC?
Line 91: The description of the PEG precipitation is not clear. What was added to the inactivated virus preparation and at what concentration?
Line 132: I assume this was done with the resuspended virus pellets from line 96? Then the material should not be referred to as "infection supernatant".
Line 122: "BEA" is not defined.
Line 141: Section 2.3 does not describe antigens, but a virus titration method. It is not clear from the manuscript what the vaccine has been prepared with. The PEG precipitate from section 2.2? With or without any further treatment or purification?
Line 148: The immunization procedure is not described.
Line 152: Please provide more detail. Did these pigs die before or after challenge? What efforts were made to determine the cause of death?
Line 168: LFPK cells are not defined.
Author Response
There is a lot of very instructive applied FMD vaccine research coming out of South Korea lately. This manuscript is another good example. It may not be highly innovative, but it is solid work that can be very useful to the field. However, the description of the methods could be improved.
Line 52: This is quite vague. There are many kinds of BHK-21 suspension cells. Which cell line was used? Is it available from a public repository such as ATCC or ECACC?
≫ We revised the manuscript according to the reviewer’s comments as follows (line 52)
The BHK-21 suspension cells that was developed from the original adherent cell line, BHK-21 [C-13] (ATCC, Manassas, VA, USA), by the Animal and Plant Quarantine Agency (APQA) and the Korea Research Institute of Bioscience & Biotechnology for use in suspension culture with serum-free media, was adapted for growth in CellventoTM BHK-200 cell culture medium (Merck, Darmstadt, Germany) by incubation at 110 rpm in a shaking incubator at 37°C with 5% CO2.
Line 91: The description of the PEG precipitation is not clear. What was added to the inactivated virus preparation and at what concentration?
≫ We revised the manuscript according to the reviewer’s comments as follows (line 94)
The inactivated viruses were treated with 50% Polyethylene Glycol (PEG) 6000 (Sigma-Aldrich) solution into a final 7.5% concentration of PEG. The solution was added with NaCl (0.5M) and then incubated for 16 h at 4℃.
Line 132: I assume this was done with the resuspended virus pellets from line 96? Then the material should not be referred to as "infection supernatant".
≫ We revised the manuscript according to the reviewer’s comments as follows (line 135)
The inactivated antigen, concentrated using Polyethylene Glycol (PEG), was carefully….
Line 122: "BEA" is not defined.
≫ We revised the manuscript according to the reviewer’s comments as follows (line 125)
BEI was created by dissolving BEA (bromoethylamine hydrobromide) in a solution….
Line 141: Section 2.3 does not describe antigens, but a virus titration method. It is not clear from the manuscript what the vaccine has been prepared with. The PEG precipitate from section 2.2? With or without any further treatment or purification?
≫ We revised the manuscript according to the reviewer’s comments (line 145)
The antigens PA-2 and A22 IRQ, as described in Section 2.2 without further purification, were mixed with 1% saponin (Sigma-Aldrich) and 10% aluminum hydroxide gel (General Chemical, NJ, USA) to prepare a monovalent vaccine.
Line 148: The immunization procedure is not described.
≫ We revised the manuscript according to the reviewer’s comments (line 154)
Each group of pigs was administered intramuscularly with vaccine containing various payload of antigens followed by homologous virus challenge at 28 day post vaccination (dpv).
Line 152: Please provide more detail. Did these pigs die before or after challenge? What efforts were made to determine the cause of death?
≫ The pig #13 died just after virus challenge and the pig #86 died at 15 days post vaccination. Necropsy revealed that the pig #`13 died of Glasser’s disease and the pig #86 died of pleuropneumonia disease.
Line 168: LFPK cells are not defined.
≫ We revised the manuscript according to the reviewer’s comments (line 175)
LFPK(fetal porcine kidney cells; supplied by Plum Island Animal Disease Center, Orient, NY, USA) cells (0.5 × 106 cells/mL) were added to….
Reviewer 2 Report
The paper “Production of Foot-and-Mouth Disease Type O and A Vaccine 2 Antigens on a Pilot-Scale and Determination of Optimal 3 Amount of Antigen for Monovalent Vaccines” describes the upscaling of a vaccine production method for serotype O and A FMDV vaccines and the testing of the different antigen doses in challenge studies in pigs.
My comments, questions and suggestions are as follows.
· In section 2.9 it states each experiment was done in triplicate. Do you mean the animal studies were done three times? Or the VN tests? The data provided show only one animal study, but they also only show one VN titre, not a mean.
· · Were all of these studies done with the same batch of Ag/vaccine? As you already know from your small-scale culture studies that the vaccine works, perhaps it would have been better to produce 3 batches at this larger scale and check for reproducible yields and in vitro properties, and compare batches or do a true dose response study in vivo.
· · Line 149: The three doses of vaccine used seem very similar. Why were these chosen as opposed to at least 3-fold or more differences? What differences were you expecting to see with such similar doses? This is not really explained well in the discussion.
· · What quantification was done on the inactivated virus particles? It seems 146S was only looked at in the cell culture supernatant. I couldn’t see where you quantified the ug of 146S in the inactivated Ag.
· · Line 152: was necropsy done? Is there any veterinary assessment to speculate at the cause of death?
· · Line 157: Do you mean a score of 1 per foot with lesions? So you also included the foot inoculated in your scoring? And therefore from your results are you saying none of the pigs developed lesions at the site of inoculation?
· · Paragraph from Line 270: Please better explain why you think myristoylation is linked to the observed differences in Ag (I assume you refer to your 146S particle measurement). Are you saying that in the A virus, stability of these may be affected by incomplete cleavage of VP0? Why this as opposed to general differences in stability due to the amino acid composition of the capsid proteins?
· · Why were no tests done for antibodies to NSPs?
Textual changes:
- Line 14: The term “highly protective” seems superfluous. Perhaps better to say the vaccine is protective against a wide range of serotype O FMDV strains in East Asia.
- Line 35: It is not clear why it is written “many countries of the WOAH” – surely it is the case that many countries world-wide have adopted FMD vaccination policies and their membership to WOAH is not relevant. Also, I don’t feel reference 5 needs to be cited here.
- Line 40: better to say “is highly efficacious against several…”
- Line 43: comma not needed after O PA-2
- Line 44: please use O PA-2, and throughout for consistency (e.g. space needed on line 68, O missing line 141)
- Line 65: ‘or’ not ‘and’
- Line 141: Section 2.3 describes VT, do you mean section 2.2?
- Line 156: perhaps better to say ‘to prevent further virus challenge to the pigs not showing clinical signs’ or something like that.
- Line 159: ‘was defined as’ is perhaps better than ‘means that’
- Line 161: This sentence seems redundant as you have just described the scoring process. You could perhaps just add the max score and the reference in that part.
- Line 251: It is not ideal to say “and has a wide range of vaccine matching”. Do you want to say it has been shown to protect against challenge with many heterologous A strains, or that in in vitro vaccine matching studies?? Please clarify this.
- Line 252: Is ‘recombinantly engineered’ an actual term? I would change to simply ‘recombinant A22 IRQ was produced for use as a vaccine strain’.
- Line 257: I feel it is an overstatement to say FMDV Ag is produced by the ton. Please rephrase.
- Line 288: This paragraph seems a repeat of the methods and results and does not really add anything to the discussion.
Author Response
In section 2.9 it states each experiment was done in triplicate. Do you mean the animal studies were done three times? Or the VN tests? The data provided show only one animal study, but they also only show one VN titre, not a mean.
≫ We revised the manuscript according to the reviewer’s comments (line 181)
We produced FMD vaccine antigens in a 100L bioreactor in triplicate. Virus titration, quantification of 146S particle and inactivation kinetics were performed three times. The animal experiment and VNT was performed only once.
Were all of these studies done with the same batch of Ag/vaccine? As you already know from your small-scale culture studies that the vaccine works, perhaps it would have been better to produce 3 batches at this larger scale and check for reproducible yields and in vitro properties, and compare batches or do a true dose response study in vivo.
≫ We produced three batches of antigens at this larger scale and carried out study with the same batch of antigen for evaluation of trial vaccine in pigs. However, the animal experiment was performed only once.
Line 149: The three doses of vaccine used seem very similar. Why were these chosen as opposed to at least 3-fold or more differences? What differences were you expecting to see with such similar doses? This is not really explained well in the discussion.
≫ Previous studies reported that antigen payload per dose was 15 ug in animal studies in pigs. However, considering the large-scale production in vaccine factories, it is economical to reduce the antigen amount by a small amount, so we adjusted the antigen payload per dose around 10 µg to 20 µg for O PA-2 and 5 µg to 15 µg for A22 IRQ for the determination of optimal antigen payload.
What quantification was done on the inactivated virus particles? It seems 146S was only looked at in the cell culture supernatant. I couldn’t see where you quantified the ug of 146S in the inactivated Ag.
≫ FMDV infection supernatant was inactivated with BEI (PA-2 and A22 IRQ was 2.64 ± 0.66 ug/ml, 1.08 ± 0.15 ug/ml, respectively) and concentrated with PEG to a final 7.5% as described in Section 2.2. The total amount of 146S in the PEG-concentrated FMDV was measured and then diluted in appropriate ratios to vary the payload per dose and used to prepare the trial vaccines.
Line 152: was necropsy done? Is there any veterinary assessment to speculate at the cause of death?
≫ The pig #13 died just after virus challenge and the pig #86 died at 15 days post vaccination. Necropsy revealed that the pig #`13 died of Glasser’s disease and the pig #86 died of pleuropneumonia disease.
Line 157: Do you mean a score of 1 per foot with lesions? So you also included the foot inoculated in your scoring? And therefore from your results are you saying none of the pigs developed lesions at the site of inoculation?
≫ When calculating the clinical score, we tallied the number of foot lesions, excluding virus inoculation site. Among the three groups that received vaccination, no lesions were observed in any area apart from the heel bulb. Conversely, in the control group that did not receive vaccination, clinical symptoms were observed in locations other than the heel bulb.
Paragraph from Line 270: Please better explain why you think myristoylation is linked to the observed differences in Ag (I assume you refer to your 146S particle measurement). Are you saying that in the A virus, stability of these may be affected by incomplete cleavage of VP0? Why this as opposed to general differences in stability due to the amino acid composition of the capsid proteins?
≫ Myristoylation of the VP4 N-terminus is known to be involved in the mature cleavage of VP0 into VP4 and VP2 during the assembly of the viral particle 146S, which allows RNA to enter the viral particle. Our assumption is based on the previous report [PLOS Pathogens 2018, 14(8)] that an inhibition of N-myistoyltranferase led to an about tenfold lower yield of 146S particle and increase of VP0 compared to the control group (without inhibitor treatment)
Why were no tests done for antibodies to NSPs?
≫ The WOAH manual states that testing for the presence of NSP antibodies in the sera of vaccinated animal is done by inoculating cattle with the vaccine for the test of vaccine purity. Since this study was carried out in pigs, we did not test for NSP antibodies. In general, pigs develop less NSP antibodies than cattle by multiple vaccination. Other study have shown that cattle did not induced NSP antibodies even though they were administered with a trial bivalent vaccine containing O Boeun and A Yeoncheon three times for the test of vaccine purity.
Line 14: The term “highly protective” seems superfluous. Perhaps better to say the vaccine is protective against a wide range of serotype O FMDV strains in East Asia.
≫ We revised the manuscript according to the reviewer’s comments (line 14)
Line 35: It is not clear why it is written “many countries of the WOAH” – surely it is the case that many countries world-wide have adopted FMD vaccination policies and their membership to WOAH is not relevant. Also, I don’t feel reference 5 needs to be cited here.
≫ We revised the manuscript according to the reviewer’s comments (line 36)
Line 40: better to say “is highly efficacious against several…”
≫ We revised the manuscript according to the reviewer’s comments (line 41)
Line 43: comma not needed after O PA-2
≫ We revised the manuscript according to the reviewer’s comments (line 44)
Line 44: please use O PA-2, and throughout for consistency (e.g. space needed on line 68, O missing line 141)
≫ We revised the manuscript according to the reviewer’s comments (line 45, 68, 145)
Line 65: ‘or’ not ‘and’
≫ We revised the manuscript according to the reviewer’s comments (line 67)
Line 141: Section 2.3 describes VT, do you mean section 2.2?
≫ We revised the manuscript according to the reviewer’s comments (line 145)
Line 156: perhaps better to say ‘to prevent further virus challenge to the pigs not showing clinical signs’ or something like that.
≫ We revised the manuscript according to the reviewer’s comments (line 162)
Line 159: ‘was defined as’ is perhaps better than ‘means that’
≫ We revised the manuscript according to the reviewer’s comments (line 166)
Line 161: This sentence seems redundant as you have just described the scoring process. You could perhaps just add the max score and the reference in that part.
≫ We revised the manuscript according to the reviewer’s comments (line 169)
Line 251: It is not ideal to say “and has a wide range of vaccine matching”. Do you want to say it has been shown to protect against challenge with many heterologous A strains, or that in in vitro vaccine matching studies?? Please clarify this.
≫ We revised the manuscript according to the reviewer’s comments (line 263)
Line 252: Is ‘recombinantly engineered’ an actual term? I would change to simply ‘recombinant A22 IRQ was produced for use as a vaccine strain’.
≫ We revised the manuscript according to the reviewer’s comments (line 264)
Line 257: I feel it is an overstatement to say FMDV Ag is produced by the ton. Please rephrase.
≫ We revised the manuscript according to the reviewer’s comments (line 269)
Generally, FMD vaccine factories manufacture FMD vaccine antigen on a large scale using multiple bioreactors
Line 288: This paragraph seems a repeat of the methods and results and does not really add anything to the discussion.
≫ We deleted the sentence according to the reviewer’s comments (line 302)
Round 2
Reviewer 2 Report
I still think the VNTs should have been performed in at least duplicate but the modifications have improved the paper overall.
The English isnt perfect but no glaring problems (and its not my job to re-write the paper :-)).
Author Response
Reviewer's comments:
I still think the VNTs should have been performed in at least duplicate but the modifications have improved the paper overall.
Response to reviewer
>> All sera samples for VNT were tested in duplicate.